# Gelating Abilities of Two-Component System of Catecholic Derivatives and a Boronic Acid

**DOI:** 10.3390/gels5040045

**Published:** 2019-10-22

**Authors:** Akihiko Tsuge, Ryota Kamoto, Daisuke Yakeya, Koji Araki

**Affiliations:** Department of Applied Chemistry, Kyushu Institute of Technology, Kitakyushu 804-8550, Japan; ryotaiueo@gmail.com (R.K.); keslted@yahoo.co.jp (D.Y.); araki@che.kyutech.ac.jp (K.A.)

**Keywords:** catechol, organogel, boronate ester

## Abstract

In the last two decades, various kinds of the low-molecular-weight organogelators (LMOGs) have been investigated in terms of technological applications in various fields as well as their fundamental scientific properties. The process of gelation is generally considered to arise from immobilization of the solvents in the three-dimensional networks formed by the assembly of gelator molecules through weak intermolecular noncovalent interactions. From these points of view a huge number of organogelators have been developed so far. In the course of our research on LMOGs we have noticed a mixture of two gelators could show a different trend in gelation compared to the single gelator. It is well known that the catecholic moiety easily forms cyclic boronate esters with the boronic acid. Thus, we have investigated the two-component system based on cyclic boronate esters formed by the catechols and a boronic acid in terms of the control of gelation capability. Basic gelation properties of the constituent catecholic gelators have also been clarified. The catecholic gelators with the amide unit form no gel by addition of the boronic acid. In contrast, the catecholic gelators with the glutamic acid moiety improve their gelation abilities by mixing with the boronic acid. Furthermore, the gelation ability of the catecholic gelators having the urea unit is maintained after addition of the boronic acid. It has been found that gelation abilities of the catecholic gelators are highly affected by addition of the boronic acid. In terms of practical applications some gels can be obtained by on-site mixture of two kinds of solutions.

## 1. Introduction

Self-assembly plays an important role in the creation of attractive functional soft materials [1]. Supramolecular gels formed by self-assembly of the low-molecular-weight organogelators (LMOGs) [2,3,4,5,6,7,8,9,10,11,12,13,14,15] are an emerging field because they are not only related to fundamental scientific interests, but also applied to a practical usage in various areas.

Although it seems difficult to predict the whole structural requirements for molecules to show gelation properties in organic solvents, the information obtained from a wide variety of organogelators helps to rationalize what kinds of chemical structures are necessary for building organogels. These required structures are closely related to non-covalent bonding such as hydrogen bonds, electrostatic attraction, hydrophobic interaction, and π–π interactions. In the last few decades a large number of organogelators have been discussed in terms of structural features, effects of noncovalent interactions, properties of gels, and aspects of their functions [16,17,18,19,20,21,22]. From this point of view we have developed organogelators based on the cyclophane skeletons [23,24], the europium complex [25], coumarin structure [26], and ferrocene unit [27].

When considering the structural unit for building a gelator molecule, the catechol group has to be attractive because non-covalent bonding, such as hydrophobic interaction, electrostatic interaction, hydrogen bonds, π–π interactions, and metal-ligand coordination bonding, can be expected, and is also well known to exhibit adhesive characteristics on a variety of surfaces [28,29,30]. Furthermore, a diol and boronic acid complex [31], which is easily formed from each component, is known as a useful tool for the construction of organized architectures in supramolecular chemistry [32,33]. This system based on a dynamic covalent bonding might be advantageous for creation of chemical stimuli-responsive soft materials [34,35,36]. In terms of gel systems using this diol/boronic acid complex, the gel materials responding to some sugars such as a glucose have been reported [37,38]. This complex has also been applied to polymer gels [39], star-shaped gels [40], and bola-shaped gels [41].

Thus, from these points of view it is expected that the gelators containing the catechol unit could provide a unique system.

Here we have designed some catecholic gelators which have the amide unit or the urea unit as the connection with the long alkyl chain and examined their preliminary gelating properties. Successively, the two-component system based on cyclic boronate esters formed by the catechols and the boronic acid has also been investigated in terms of formation of gels.

## 2. Results and Discussion

We have synthesized three kinds of the catecholic derivatives (**2**, **4**, **6**) as shown in Scheme 1. The catecholic derivatives **2a**–**d** were obtained by treating **1a**–**d** prepared from 3,4-dimethoxybenzoyl chloride and the corresponding alkylamines with BBr_3_ in the yields of 44–74%. The reaction of 3,4-dimethoxybenzoic acid and the corresponding alkylamines in the presence of diphenylphosphoryl azide (DPPA) in DME gave the urea compounds **3a**–**d**, followed by demethylation with BBr_3_ to afford the desired catecholic derivatives **4a**–**d** in the yields of 74–94%. The compounds **5a**–**d** were synthesized by condensation of the corresponding glutamic acid derivatives and 3,4-dimethoxybenzoic acid using diethyl cyanophosphonate (DECP). By treating **5a**–**d** with BBr_3_ the catecholic derivatives having the glutamic acid moiety **6a**–**d** were obtained in the yields of 50–76%. The referential compounds **7** and **8** were also prepared by the similar method as described above. The boronic acid carrying the long alkyl chain **9** was obtained by the condensation of 4-carboxyphenylboronic acid and the corresponding alkylamine under DECP in the yields of 74% as shown in Scheme 1.

We have examined various kinds of solvents for the gelation behaviors of the catecholic derivatives prepared here. The catecholic derivatives **2a**–**d** show very good solubility against polar solvents such as DMSO, DMF, EtOH, and MeOH as in Table 1, resulting in no gelation. The gelation properties of **2a**–**d** in other solvents are also summarized in Table 1. Formation of gel in toluene and benzene was confirmed for **2c** and **2d**. It was found out that the compound **2d** having the longer alkyl chain gelated chloroform.

The gelation properties of the catecholic derivatives **4a**–**d** were also examined in various solvents as summarized in Table 1. **4b**,**c** show a different trend of gelation between toluene and benzene. Although a stable gel from **4b** and **4c** in toluene was observed, a partial gel was only formed in benzene. The partial gel means that the gel once formed on cooling tends to collapse after a while. Chloroform can be effectively gelated by **4d**. The only compound that can gelate EtOH is **4d**, regardless of excellent solubility of **4a**–**d** in alcohols. The compound **4d** with the long alkyl chain tends to form the gel in DMF and DMSO. The optical image and SEM analysis of the gel from **4d** in toluene are shown in Figure 1a as an example. A developed network of elongated fibers was observed.

The catecholic derivatives **6a**–**d** exhibit a poor solubility in hexane and cyclohexane, and an excellent solubility in DMF and DMSO. Some gel formations were confirmed as shown in Table 2. Compound **6d** is able to gelate toluene and benzene. Interestingly, **6c** tends to build a partial gel in these solvents. As an example the optical image and SEM analysis of the gel from **6d** in DMSO are shown in Figure 1b. Relatively thick fibers were observed.

In order to know the effect of the hydroxyl group we examined the gelation properties of the phenol derivative **7** as well as the compound **8**. The results are also summarized in Table 2.

Although no gelation from **7** in any solvents examined was observed, the compound **8** carrying three hydroxy groups clearly shows better gelation abilities compared to **2c**. The minimum concentration of **8** for gelation in toluene and benzene is about ten times lower than that of **2c**. This result strongly indicates that the hydroxy group plays a crucial role in gel formation.

Although the boronic acid **9** tends to dissolve in EtOH, MeOH, DMF, and DMSO, its precipitate was produced in solvents such as benzene, toluene, and chloroform. Thus, the boronic acid **9** itself showed no gelation in organic solvents examined. We carried out the gelation test for the 1:1 mixture of catecholic compounds and **9** as summarized in Table 3. The 1:1 mixture of **2c** and **9** in toluene and benzene shows no gelation despite several trials. The similar results were obtained for the mixture of **2d**. Although the gel is formed by **2d** in chloroform, the 1:1 mixture with **9** cannot produce such a gel. This decline of the ability in gelation can be ascribed to the loss of the hydroxy groups due to formation of a boronate ester.

In contrast it has been found out that **4b**–**d** maintain their ability for gelation in toluene and chloroform even after mixing with **9** as shown in Table 3. It is interesting that the stable gels of **4b** and **4c** were formed in benzene by mixing with **9** because they form the partial gels without **9**. In this case the urea group might compensate for the loss of hydrogen bonding based on the hydroxy groups. We have examined thermal stability of the gels from **4d** and **4d**/**9** in chloroform. The gels from both of them collapsed at the same temperature (ca. 45 °C).

On the other hand the improvement of gelation ability for the compounds **6** can be also observed when they are mixed with **9** (Table 3). Although no change for gelation properties of **6b** was observed, some improvements in gelation were confirmed for **6c** and **6d**. Compound **6c** itself exhibited a partial gel in benzene and toluene, however, the stable gel was formed by the 1:1 mixture with **9**. Furthermore the mixture of **6d** also shows formation of gel in chloroform despite no gelation by **6d** itself.

Figure 2 shows the optical image and SEM analysis of the gel from the mixture of **6d** and **9** in chloroform. A clear gel was obtained and a developed network was observed. The reason for this enhancement of gelation is still unclear, however, it is expected that three long alkyl chains in the molecule could contribute to an effective molecular assembly.

## 3. Conclusions

We have prepared organogelators based on the catechol unit having the amide, the urea, and glutamic acid moieties. We have clarified their basic properties of gelation in organic solvents. It has also been found that the gelation abilities of these catecholic gelators are highly affected by the addition of a boronic acid. The catecholic gelators with the amide unit lose their gelation abilities by the addition of a boronic acid. No effect on gelation properties was observed for the catecholic gelators with the urea unit. In contrast it should be noted that the catecholic gelators with the glutamic acid moiety improve their gelation abilities by mixing with the boronic acid. These phenomena might be ascribed to formation of the boronic ester. Further research according to this concept is currently underway in our laboratory. It should be noted that the system described here has an advantage from the viewpoint of practical usage because the gel can be formed on-site by mixing up two kinds of solutions containing the catecholic derivative and the boronic acid, respectively.

## 4. Materials and Methods

### 4.1. General

All chemicals were purchased from commercial suppliers and used without further purification. Melting points were obtained by a Yanagimoto MP-52 melting point apparatus (AE-MIC Trading, Kyoto, Japan). Nuclear magnetic resonance (NMR) spectra were measured on a Bruker ADVANCE HD 500 spectrometer (Billerica, MA, USA) with Me_4_Si as the internal reference. *J* values are given in Hz. Mass spectra were recorded by using a JEOL JMS-SX102A spectrometer (JEOL, Tokyo, Japan). Elemental analysis was carried out with Yanaco MT-6 CHN recorder (Yanaco New Science Inc., Kyoto, Japan).

### 4.2. Typical Procedure for Syntheses of ***2a***–***d***

After a solution of 3,4-dimethoxybenzoyl chloride (4.40 g, 22 mmol) and hexylamine (2.92 mL, 22 mmol) together with triethylamine (3.20 mL, 23 mmol) in chloroform (50 mL) was stirred for 2 h at room temperature, the reaction mixture was washed with brine and water, then dried over sodium sulfate. The chloroform solution was evaporated under reduced pressure to leave a residue that was recrystallized from hexane to give **1a** (4.90 g, 84%). To a stirred solution of **1a** (2.01 g, 7.6 mmol) in dichloromethane (40 mL) at 0 °C a solution of BBr_3_ (4.72 g, 18.8 mmol) in dichloromethane (40 mL) was added dropwise. After the reaction mixture was stirred for 13 h at room temperature, it was poured into ice-water. The mixture was stirred for a further 2 h at room temperature and extracted with ethyl acetate. The extract was washed with water, dried over sodium sulfate, and evaporated under reduced pressure to afford a residue which, on recrystallization from ethyl acetate and hexane, gave **2a** as white powder (1.18 g, 65%).

*Compound***2a***:* Yield 65%; white powder; mp 152–154 °C (EtOAc/hexane); IR ν_max_ (KBr)/cm^−1^: 3328, 3045, 2960, 1642, 1584, 1495, 1425, 1364, 1226, 1050, 1022, 805, 685; ^1^H NMR (500 MHz; DMSO-*d*_6_) δ 0.86 (3H, t, *J* = 6.8 CH_3_), 1.26–1.32 (6H, m, CH_2_), 1.46–1.50 (2H, m, CH_2_), 3.18–3.20 (2H, m, CH_2_NH), 6.73 (1H, d, *J* = 7.0 ArH), 7.16 (1H, dd, *J* = 1.6, 7.0 ArH), 7.26 (1H, d, *J* = 1.6 ArH), 8.10 (1H, s, NH), 9.05 (1H, s, ArOH), 9.40 (1H, s, ArOH); ^13^C NMR (125 MHz, DMSO-*d*_6_) δ 14.2, 23.4, 27.0, 31.8, 32.2, 68.4, 118.2, 120.2, 128.4, 132.8, 151.6, 152.4, 170.6; fast atom bombardment (FAB) MS *m*/*z* 237 (M^+^); Anal. Found: C, 65.64; H, 8.33; N, 5.75%, Calc. for C_13_H_19_NO_3_: C, 65.79; H, 8.09; N, 5.90%. 

*Compound***2b**: Yield 69%; white powder; mp 140–143 °C (EtOAc/hexane); IR ν_max_ (KBr)/cm^−1^: 3330, 3050, 2960, 1642, 1580, 1495, 1425, 1360, 1226, 1052, 1022, 805, 746, 685; ^1^H NMR (500 MHz; DMSO-*d*_6_) δ 0.86 (3H, t, *J* = 6.8 CH_3_), 1.26–1.30 (10H, m, CH_2_), 1.46–1.48 (2H, m, CH_2_), 3.16–3.20 (2H, m, CH_2_NH), 6.73 (1H, d, *J* = 7.0 ArH), 7.16 (1H, dd, *J* = 1.6, 7.0 ArH), 7.26 (1H, d, *J* = 1.6 ArH), 8.09 (1H, s, NH), 9.07 (1H, s, ArOH), 9.40 (1H, s, ArOH); ^13^C NMR (125 MHz, DMSO-*d*_6_) δ 14.2, 23.1, 27.4, 29.4, 30.4, 32.4, 68.4, 118.2, 120.0, 128.4, 132.8, 151.8, 152.4, 170.8; FAB MS *m*/*z* 265 (M^+^); Anal. Found: C, 67.61; H, 8.82; N, 5.44%, Calc. for C_15_H_23_NO_3_: C, 67.88; H, 8.75; N, 5.28%.

*Compound***2c**: Yield 74%; white powder; mp 142–145 °C (EtOAc/hexane); IR ν_max_ (KBr)/cm^−1^: 3332, 3045, 2960, 1640, 1580, 1490, 1430, 1360, 1224, 1052, 1028, 805, 746, 685; ^1^H NMR (500 MHz; DMSO-*d*_6_) δ 0.85 (3H, t, *J* = 6.8 CH_3_), 1.22–1.28 (14H, m, CH_2_), 1.44–1.48 (2H, m, CH_2_), 3.18–3.20 (2H, m, CH_2_NH), 6.74 (1H, d, *J* = 7.0 ArH), 7.16 (1H, dd, *J* = 1.6, 7.0 ArH), 7.22 (1H, d, *J* = 1.6 ArH), 8.09 (1H, s, NH), 9.04 (1H, s, ArOH), 9.40 (1H, s, ArOH); ^13^C NMR (125 MHz, DMSO-*d*_6_) δ 14.2, 23.1, 27.6, 28.0, 29.4, 30.0, 30.4, 31.6, 68.4, 118.2, 121.0, 128.4, 132.8, 151.8, 152.4, 171.0; FAB MS *m*/*z* 293 (M^+^); Anal. Found: C, 69.41; H, 9.44; N, 4.94%, Calc. for C_17_H_27_NO_3_: C, 69.58; H, 9.29; N, 4.77%.

*Compound***2d**: Yield 44%; white powder; mp 122–125 °C (EtOAc/hexane); IR ν_max_ (KBr)/cm^−1^: 3328, 3040, 2965, 1640, 1580, 1495, 1425, 1360, 1224, 1092, 1028, 805, 734, 685; ^1^H NMR (500 MHz; DMSO-*d*_6_) δ 0.85 (3H, t, *J* = 6.8 CH_3_), 1.22–1.32 (18H, m, CH_2_), 1.44–1.46 (2H, m, CH_2_), 3.18–3.22 (2H, m, CH_2_NH), 6.74 (1H, d, *J* = 7.0 ArH), 7.18 (1H, dd, *J* = 1.6, 7.0 ArH), 7.22 (1H, d, *J* = 1.6 ArH), 8.09 (1H, s, NH), 9.04 (1H, s, ArOH), 9.42 (1H, s, ArOH); ^13^C NMR (125 MHz, DMSO-*d*_6_) δ 14.2, 23.1, 27.8, 28.4, 29.4, 30.4, 30.8, 31.6, 68.4, 118.2, 122.0, 128.4, 131.8, 152.0, 152.4, 171.0; FAB MS *m*/*z* 321 (M^+^); Anal. Found: C, 70.77; H, 9.88; N, 4.50%, Calc. for C_19_H_31_NO_3_: C, 70.97; H, 9.74; N, 4.36%.

### 4.3. Typical Procedure for Syntheses of ***4a***–***d***

After a solution of 3,4-dimethoxybenzoic acid (1.50 g, 8.23 mmol) and diphenylphosphoryl azide (DPPA) (1.88 mL, 8.72 mmol) together with triethylamine (1.65 mL, 11.9 mmol) in 1,2-dimethoxyethane (DME) (50 mL) was stirred for 3 h at room temperature, hexylamine (1.10 mL, 8.32 mmol) was added, followed by reflux for 1 h, and then extracted by dichloromethane. The extract was washed with water, dried over sodium sulfate, followed by evaporation under reduced pressure to afford a residue, which was washed with hexane to give **3a** (1.04 g, 45%). To a stirred solution of 1-hexyl-3-[3,4-(dimethoxy)phenyl]urea (1.00 g, 3.57 mmol) in CH_2_Cl_2_ (40 mL) at 0 °C a solution of BBr_3_ (2.23 g, 8.90 mmol) in CH_2_Cl_2_ (40 mL) was added dropwise. After the reaction mixture was stirred for 13 h at room temperature, it was poured into ice-water. The mixture was stirred for a further 2 h at room temperature and extracted with dichloromethane. This dichloromethane extract was washed with water, dried over sodium sulfate, and evaporated under reduced pressure to yield a residue which, on recrystallization from MeOH and water, gave **4a** as white powder (0.72 g, 79%).

*Compound***4a**: Yield 79%; white powder; mp 160–162 °C (MeOH/water); IR ν_max_ (KBr)/cm^−1^: 3330, 3045, 2960, 1650, 1584, 1495, 1425, 1364, 1226, 1050, 1022, 815, 695; ^1^H NMR (500 MHz; DMSO-*d*_6_) δ 0.86 (3H, t, *J* = 6.8 CH_3_), 1.24–1.28 (6H, m, CH_2_), 1.36–1.38 (2H, m, CH_2_), 2.98–3.04 (2H, m, CH_2_NH), 5.87–5.89 (1H, brs, NH), 6.51 (1H, dd, *J* = 1.6, 7.6 ArH), 6.55 (1H, d, *J* = 7.6 ArH), 6.91 (1H, d, *J* = 1.6 ArH), 7.93 (1H, s, NH), 8.32 (1H, s, ArOH), 8.78 (1H, s, ArOH); ^13^C NMR (125 MHz, DMSO-*d*_6_) δ 14.2, 23.1, 26.4, 31.7, 31.7, 69.6, 118.4, 120.2, 128.0, 132.6, 152.6, 153.4, 168.6; FAB MS *m*/*z* 252 (M^+^); Anal. Found: C, 61.62; H, 8.21; N, 11.05%, Calc. for C_13_H_20_N_2_O_3_: C, 61.87; H, 8.00; N, 11.10%.

*Compound***4b**: Yield 86%; white powder; mp 152–155 °C (MeOH/water); IR ν_max_ (KBr)/cm^−1^: 3334, 3045, 2960, 1650, 1584, 1498, 1425, 1360, 1226, 1050, 1020, 815, 695, 650; ^1^H NMR (500 MHz; DMSO-*d*_6_) δ 0.86 (3H, t, *J* = 6.8 CH_3_), 1.24–1.32 (10H, m, CH_2_), 1.38–1.40 (2H, m, CH_2_), 3.00–3.04 (2H, m, CH_2_NH), 5.87–5.90 (1H, brs, NH), 6.50 (1H, dd, *J* = 1.6, 7.6 ArH), 6.55 (1H, d, *J* = 7.6 ArH), 6.90 (1H, d, *J* = 1.6 ArH), 7.93 (1H, s, NH), 8.32 (1H, s, ArOH), 8.78 (1H, s, ArOH); ^13^C NMR (125 MHz, DMSO-*d*_6_) δ 14.2, 23.2, 27.4, 29.4, 30.0, 32.4, 69.4, 118.2, 120.0, 128.4, 132.8, 152.4, 153.4, 168.6; FAB MS *m*/*z* 280 (M^+^); Anal. Found: C, 64.13; H, 8.81; N, 9.77%, Calc. for C_15_H_24_N_2_O_3_: C, 64.25; H, 8.64; N, 9.99%.

*Compound***4c**: Yield 94%; white powder; mp 148–152 °C (MeOH/water); IR ν_max_ (KBr)/cm^−1^: 3330, 3045, 2965, 1650, 1584, 1500, 1425, 1356, 1226, 1050, 1020, 810, 695, 650; ^1^H NMR (500 MHz; DMSO-*d*_6_) δ 0.86 (3H, t, *J* = 6.8 CH_3_), 1.24–1.34 (14H, m, CH_2_), 1.38–1.40 (2H, m, CH_2_), 2.98–3.04 (2H, m, CH_2_NH), 5.87–5.92 (1H, brs, NH), 6.50 (1H, dd, *J* = 1.6, 7.8 ArH), 6.55 (1H, d, *J* = 7.8 ArH), 6.90 (1H, d, *J* = 1.6 ArH), 7.94 (1H, s, NH), 8.32 (1H, s, ArOH), 8.78 (1H, s, ArOH); ^13^C NMR (125 MHz, DMSO-*d*_6_) δ 14.2, 23.0, 27.6, 28.0, 29.4, 30.2, 30.4, 31.6, 68.4, 118.2, 121.0, 128.4, 132.8, 152.6, 153.6, 168.6; FAB MS *m*/*z* 308 (M^+^); Anal. Found: C, 65.96; H, 9.22; N, 8.89%, Calc. for C_17_H_28_N_2_O_3_: C, 66.19; H, 9.17; N, 9.08%.

*Compound***4d**: Yield 74%; white powder; mp 130–132 °C (MeOH/water); IR ν_max_ (KBr)/cm^−1^: 3335, 3045, 2965, 1650, 1584, 1504, 1428, 1356, 1226, 1050, 1022, 810, 695; ^1^H NMR (500 MHz; DMSO-*d*_6_) δ 0.86 (3H, t, *J* = 6.8 CH_3_), 1.26–1.34 (18H, m, CH_2_), 1.36–1.38 (2H, m, CH_2_), 2.98–3.02 (2H, m, CH_2_NH), 5.87–5.90 (1H, brs, NH), 6.50 (1H, dd, *J* = 1.6, 7.6 ArH), 6.58 (1H, d, *J* = 7.6 ArH), 6.88 (1H, d, *J* = 1.6 ArH), 7.94 (1H, s, NH), 8.32 (1H, s, ArOH), 8.78 (1H, s, ArOH); ^13^C NMR (125 MHz, DMSO-*d*_6_) δ 14.2, 23.1, 27.8, 28.4, 29.4, 30.2, 30.8, 31.6, 68.4, 118.2, 122.0, 128.4, 132.0, 152.2, 153.6, 168.6; FAB MS *m*/*z* 336 (M^+^); Anal. Found: C, 67.60; H, 9.74; N, 8.18%, Calc. for C_19_H_32_N_2_O_3_: C, 67.81; H, 9.60; N, 8.33%.

### 4.4. Typical Procedure for Syntheses of ***6a***–***d***

To a solution of 3,4-dimethylbenzoic acid (0.52 g, 2.84 mmol) and l-glutamic acid derivative (*n* = 6) (0.88 g, 2.85 mmol) in the presence of triethylamine (0.66 mL, 5.03 mmol) in Tetrahydrofuran (THF) (50 mL) in an ice bath was added a solution of diethyl cyanophosphonate (DECP) (0.45 mL, 3.03 mmol) in THF (10 mL). The reaction mixture was stirred for 20 h, then the solvent was removed to leave orange solid. This solid was suspended in aqueous 10% NaOH and stirred for 1 h, followed by filtration. The resulting solid was washed by 10% HCl solution and water to give white powder. Further purification was carried out by repeated washing with hexane to afford **5a** (1.08 g, 79%). To a stirred solution of **5a** (1.21 g, 2.54 mmol) in CH_2_Cl_2_ (40 mL) at 0 °C a solution of BBr_3_ (2.50 g, 10.0 mmol) in CH_2_Cl_2_ (40 mL) was added dropwise. After the reaction mixture was stirred for 20 h at room temperature, it was poured into ice-water, and then was stirred for a further 2 h at room temperature. The mixture was extracted with ethyl acetate. The extract was washed with water, dried over sodium sulfate, and then evaporated under reduced pressure to afford a residue which, on recrystallization from ethyl acetate and hexane, gave **6a** as pale yellow powder (0.86 g, 76%).

*Compound***6a**: Yield 76%; pale yellow powder; mp 155–158 °C (EtOAc/hexane); IR ν_max_ (KBr)/cm^−1^: 3330, 3045, 2960, 1640, 1580, 1495, 1425, 1360, 1226, 805, 676; ^1^H NMR (500 MHz; DMSO-*d*_6_) δ 0.88 (3H, t, *J* = 6.8 CH_3_), 1.23–1.30 (12H, m, CH_2_), 1.48–1.52 (4H, m, CH_2_), 1.93–1.95 (2H, m, CH_2_), 2.30–2.33 (2H, m, CH_2_), 3.22–3.26 (4H, m, CH_2_NH), 4.32–4.34 (1H, m, CH), 6.32 (1H, t, *J* = 6.9 NH), 6.75 (1H, d, *J* = 7.0 ArH), 6.81 (1H, t, *J* = 6.9 NH), 7.23 (1H, dd, *J* = 1.6, 7.0 ArH), 7.30 (1H, d, *J* = 1.6 ArH), 8.15 (1H, t, *J* = 6.9 NH), 8.32 (1H, s, ArOH), 8.78 (1H, s, ArOH); ^13^C NMR (125 MHz, DMSO-*d*_6_) δ 14.0, 23.4, 27.2, 31.8, 32.2, 68.4, 118.0, 122.2, 128.4, 132.8, 151.4, 152.8, 165.0, 169.0, 171.6; FAB MS *m*/*z* 449 (M^+^); Anal. Found: C, 63.94; H, 8.93; N, 9.25%, Calc. for C_24_H_39_N_3_O_5_: C, 64.10; H, 8.76; N, 9.35%.

*Compound***6b**: Yield 50%; pale yellow powder; mp 148–152 °C (EtOAc/hexane); IR ν_max_ (KBr)/cm^−1^: 3335, 3045, 2960, 1640, 1583, 1495, 1425, 1360, 1220, 1050, 805, 674; ^1^H NMR (500 MHz; DMSO-*d*_6_) δ 0.88 (3H, t, *J* = 6.8 CH_3_), 1.23–1.28 (20H, m, CH_2_), 1.50–1.52 (4H, m, CH_2_), 1.93–1.95 (2H, m, CH_2_), 2.31–2.33 (2H, m, CH_2_), 3.24–3.26 (4H, m, CH_2_NH), 4.34–4.36 (1H, m, CH), 6.32 (1H, t, *J* = 6.9 NH), 6.75 (1H, d, *J* = 7.0 ArH), 6.81 (1H, t, *J* = 6.9 NH), 7.23 (1H, dd, *J* = 1.6, 7.0 ArH), 7.30 (1H, d, *J* = 1.6 ArH), 8.15 (1H, t, *J* = 6.9 NH), 8.26 (1H, s, ArOH), 8.72 (1H, s, ArOH); ^13^C NMR (125 MHz, DMSO-*d*_6_) δ 14.2, 23.0, 27.2, 29.4, 30.4, 32.4, 68.0, 117.8, 120.0, 128.4, 132.8, 152.8, 164.8, 169.0, 171.6; FAB MS *m*/*z* 505 (M^+^); Anal. Found: C, 66.34; H, 9.53; N, 8.26%, Calc. for C_28_H_47_N_3_O_5_: C, 66.49; H, 9.39; N, 8.31%.

*Compound***6c**: Yield 66%; pale yellow powder; mp 146–148 °C (EtOAc/hexane); IR ν_max_ (KBr)/cm^−1^: 3332, 3045, 2964, 1640, 1580, 1495, 1430, 1355, 1220, 1050, 805, 670; ^1^H NMR (500 MHz; DMSO-*d*_6_) δ 0.86 (3H, t, *J* = 6.8 CH_3_), 1.20–1.28 (28H, m, CH_2_), 1.50–1.52 (4H, m, CH_2_), 1.90–1.95 (2H, m, CH_2_), 2.31–2.34 (2H, m, CH_2_), 3.24–3.26 (4H, m, CH_2_NH), 4.34–4.36 (1H, m, CH), 6.55 (1H, t, *J* = 6.9 NH), 6.72 (1H, d, *J* = 7.0 ArH), 6.91 (1H, t, *J* = 6.9 NH), 7.23 (1H, dd, *J* = 1.6, 7.0 ArH), 7.30 (1H, d, *J* = 1.6 ArH), 8.15 (1H, t, *J* = 6.9 NH), 8.26 (1H, s, ArOH), 8.70 (1H, s, ArOH); ^13^C NMR (125 MHz, DMSO-*d*_6_) δ 14.2, 23.0, 27.6, 28.2, 29.4, 30.0, 30.4, 31.6, 68.4, 118.2, 121.0, 128.0, 132.8, 152.8, 164.6, 169.0, 171.5; FAB MS *m*/*z* 561 (M^+^); Anal. Found: C, 68.14; H, 9.96; N, 7.30%, Calc. for C_32_H_55_N_3_O_5_: C, 68.40; H, 9.89; N, 7.48%.

*Compound***6d**: Yield 62%; pale yellow powder; mp 130–133 °C (EtOAc/hexane); IR ν_max_ (KBr)/cm^−1^: 3335, 3045, 2964, 1640, 1580, 1498, 1430, 1355, 1220, 1050, 805, 676; ^1^H NMR (500 MHz; DMSO-*d*_6_) δ 0.88 (3H, t, *J* = 6.8 CH_3_), 1.20–1.28 (36H, m, CH_2_), 1.50–1.52 (4H, m, CH_2_), 1.92–1.95 (2H, m, CH_2_), 2.31–2.34 (2H, m, CH_2_), 3.24–3.26 (4H, m, CH_2_NH), 4.26–4.30 (1H, m, CH), 6.43 (1H, t, *J* = 6.9 NH), 6.72 (1H, d, *J* = 7.0 ArH), 6.90 (1H, t, *J* = 6.9 NH), 7.23 (1H, dd, *J* = 1.6, 7.0 ArH), 7.32 (1H, d, *J* = 1.6 ArH), 8.15 (1H, t, *J* = 6.9 NH), 8.26 (1H, s, ArOH), 8.70 (1H, s, ArOH); ^13^C NMR (125 MHz, DMSO-*d*_6_) δ 14.2, 23.1, 27.6, 28.4, 29.4, 30.4, 30.6, 31.6, 68.4, 118.2, 122.2, 128.4, 131.8, 151.4, 152.8, 164.6, 169.0, 171.5; FAB MS *m*/*z* 617 (M^+^); Anal. Found: C, 69.87; H, 10.45; N, 6.72%, Calc. for C_36_H_63_N_3_O_5_: C, 69.96; H, 10.30; N, 6.80%.

### 4.5. Procedure for Synthesis of ***9***

To a solution of 4-carboxyphenylboroic acid (1.00 g, 6.03 mmol) and decylamine (0.83 mL, 6.30 mmol) together with triethylamine (1.30 mL, 10.0 mmol) in Dimethylformamide (DMF) (50 mL) in an ice bath was added a solution of diethyl cyanophosphonate (DECP) (1.10 mL, 7.24 mmol) in DMF (20 mL). After the reaction mixture was stirred for 24 h, the solvent was removed to leave the solid. This solid was suspended in aqueous 10% NaOH and stirred for 1 h, followed by filtration. The resulting solid was washed by 10% HCl solution and water to give white powder. Further purification was done by recrystallization with hexane to afford **9** as white powder (1.20 g, 74%).

*Compound***9***:* Yield 74%; white powder; mp 122–126 °C (hexane); IR ν_max_ (KBr)/cm^−1^: 3335, 3045, 2964, 1642, 1584, 1495, 1425, 1364, 1226, 1050, 1020, 805, 680; ^1^H NMR (500 MHz; DMSO-*d*_6_) δ 0.85 (3H, t, *J* = 6.8 CH_3_), 1.24–1.30 (14H, m, CH_2_), 1.48–1.50 (2H, m, CH_2_), 3.20–3.22 (2H, m, CH_2_NH), 3.63 (2H, brs, OH), 7.77 (2H, d, *J* = 1.6, ArH), 7.83 (2H, d, *J* = 1.6 ArH), 8.43 (1H, t, *J* = 6.9 NH); ^13^C NMR (125 MHz, DMSO-*d*_6_) δ 14.0, 23.2, 27.8, 28.4, 29.4, 30.0, 30.8, 31.6, 68.4, 123.0, 128.4, 131.8, 142.0, 171.0; FAB MS *m*/*z* 304 (M^+^); Anal. Found: C, 66.57; H, 9.35; N, 4.40%, Calc. for C_17_H_28_BNO_3_: C, 66.88; H, 9.26; N, 4.59%.

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
