# Peer review of "Gelating Abilities of Two-Component System of Catecholic Derivatives and a Boronic Acid"

_gels, 2019, doi:10.3390/gels5040045_

Round 1
Reviewer 1 Report
This manuscript described the synthesis of low molecular weight organogelators with catechol and boronic acid moieties, and the gelation abilities of each gelator and two component gelators in various organic solvents. The organogelators contain alkyl chain(s), and amide and/or urea bonds in their skeletons. These functional groups are very common for the organogelators and similar units have been already reported. Therefore the organogel formation based on fibrous aggregates network from the small molecules in the present article is not surprisingly. The authors should clearly describe the characteristic properties and the specific functions of the organogelators designed in this paper. The authors emphasized that the two-component system based on organogelators with catechol and boronic acid moieties were highly effective on the gelation ability. However, two component organogelators have been already reported so far, and their gelation ability was generally enhanced by mixing of them. Further discussions on the two-component system will be required with more fundamental analysis, and results should be compared with others.
The authors expected specific functions based on the interaction between catecholic and boromic acid groups in the organogelators. However only gelation properties of the mixture were demonstrated. More analysis and demonstrations for their functions are needed.
In summary, I do not recommend this manuscript for the publication in the journal of “gels” in its present style. I think that further study on the functions and specific features of the two-component system will be needed.
Author Response
To Reviewer 1
Point 1: This manuscript described the synthesis of low molecular weight organogelators with catechol and boronic acid moieties, and the gelation abilities of each gelator and two component gelators in various organic solvents. The organogelators contain alkyl chain(s), and amide and/or urea bonds in their skeletons. These functional groups are very common for the organogelators and similar units have been already reported. Therefore the organogel formation based on fibrous aggregates network from the small molecules in the present article is not surprisingly.
Response 1: As reviewer 1 suggests above, the catecholic organogelators prepared here have no particular characteristics. Here we just had to describe their basic properties prior to mixing with the boronic acid, because the main purpose of this research should be comparison of gelation properties between before and after mixing the catechol-type organogelators with the boronic acid.
Point 2: The authors should clearly describe the characteristic properties and the specific functions of the organogelators designed in this paper.
Response 2: Again as the reviewer 1 has referred, it is not surprising that the catecholic organogelators described here form organogels in some organic solvents. What we would like to present here is whether we could control the gelation abilities of these catecholic organogelators by mixing of the boronic acid. As the results three different patterns for the mixture of catecholic organogelators and the boronic acid have been found out as follows:
by addition of the boronic acid the catecholic gelators (2c, 2d) lose the gelation ability, the catecholic gelators (4b, 4c, 4d) maintain or improve the gelation ability, and the catecholic gelators (6d) enhance the gelation ability. In order to clarify these facts Table 3 was revised. The results of the mixture of 2c/9 and 2d/9 were added. And some sentences were added in Abstract section and Conclusion section to emphasize our results.
Point 3: The authors emphasized that the two-component system based on organogelators with catechol and boronic acid moieties were highly effective on the gelation ability. However, two component organogelators have been already reported so far, and their gelation ability was generally enhanced by mixing of them.
Response 3: As reviewer 1 mentions above two-component systems employing some boronic acids have been reported. The following papers are mainly concerned with such a two-component system. As far as I know most of them are concerned with sugar-responsive system, polymer gels and so forth. On the other hand our present system has focused on the creation of new gel system by just mixing two components that are the catecholic compounds and the boronic acid.
According to reviewer 1’s comments, we have revised the introduction part in order to clarify our purpose based on the results of the following five reports. Here we should emphasize that a simple mixing of the catecholic gelators with the boronic acid could provide three different results for the gelation properties.
1) Influence of length and structure of aryl boronic acid crosslinkers on organogels with partially hydrolyzed poly(vinyl acetate)
[Duncan, T. T.; Weiss, R. G. Colloid Polym. Sci. 2018, 296, 1047-1056. DOI: 10.1007/s00396-018-4326-7]
Here two-component organogels from the aryl boronic acid as the crosslinker and polymer backbone have been characterized. This research deals with the polymer.
2) A novel glucose/pH responsive low-molecular-weight organogel of easy recycling
[Zhou, C.; Gao, W.; Yang, K.; Xu, L.; Ding, J.; Chen, J.; Lin, M.; Huang, X.; Wang, S.; Wu, H. Langmuir, 2013, 29, 13568-13575. DOI: 10.1021/la4033578 PubMed ID: 24093805]
Here the phenylboronic acid based gel which shows a response to glucose has been developed. This research focuses on development of some sensing gels.
3) Development of chemical stimuli-responsive organogel using boronate ester-substituted cyelotrieatechylene.
[Kubo, Y.; Yoshizumi, W.; Minami, T. Chem. Lett. 2008, 37, 1238-1239. DOI: 10.1246/cl.2008.1238]
Here two-component organogels from the boronic acid and star-shaped diol has been reported. They have used the boronate ester-substituted compounds themselves. Thus, effects of mixing of two components have not mentioned.
4) Organogel or polymer gel; facilitated gelation of a sugar-based organic gel by the addition of a boronic acid-appended polymer.
[Kobayashi, H.; Amaike, M.; Jung, J. H.; Friggeri, A.; Shinkai, S.; Resinhoudt, D. N. Chem. Commun. 2001, 1038-1039. DOI: 10.1039/b102436c]
Here the interaction of boronic acid-appended polymer with sugar-based polymer has been clarified. They have employed the polymer as a component.
5) TEM and SEM observations of super-structures constructed in organogel systems from a combination of boronic-acid-appended bola-amphiphiles with chiral diols.
[Koumoto, K.; Yamashita, T.; Kimura, T.; Luboradzki. R.; Shinkai, S.
Nanotechnology, 2001, 12, 25-31. DOI: 10.1088/0957-4484/12/1/306]
Here the structure of the two-component organogel constructed from boronic acid bola-amphiphiles and diol compounds has been examined. They have used chiral diols.
Point 4: Further discussions on the two-component system will be required with more fundamental analysis, and results should be compared with others. The authors expected specific functions based on the interaction between catecholic and boromic acid groups in the organogelators. However only gelation properties of the mixture were demonstrated.
Response 4: In this paper we believe that the most important matter is to show some results of simple mixing of the catecholic gelators with the boronic acid in terms of gelation properties. Thus, we have just shown whether gels could be formed or not after mixing two components.
Point 5: More analysis and demonstrations for their functions are needed. In summary, I do not recommend this manuscript for the publication in the journal of “gels” in its present style. I think that further study on the functions and specific features of the two-component system will be needed.
Response 4: Although I totally agree with this comment, I believe the most important initial stage for research on “gel” is to know whether the gel can be formed or not. Thus, more detailed properties and functions of the gels formed by mixing of two components should be clarified in the next stage. Again I just want to emphasize that there are three different patterns for gel formations by simple mixing of the catecholic gelators with the boronic acid.
Reviewer 2 Report
As the authors wrote scientific interest towards supramolecular organogels has been large in the past decade. Taking into account this the list of references is surprisingly short? Therefore at least the very recent review "Shaping and structuring supramolecular gels" by Chivers and Smith, Nature Rev. Mat. 4, 463 - 478 (2019) should be included.
Although this work fulfills all the basic demands of a well conducted study, more details about the thermal stability and rheological properties of the gels should be included at least for some congeners.
The authors should also provide a probable explanation why the elemental analyses fail for some compunds (no satisfactory result was obtained).
Author Response
To Reviewer 2
Point 1: As the authors wrote scientific interest towards supramolecular organogels has been large in the past decade. Taking into account this the list of references is surprisingly short? Therefore at least the very recent review "Shaping and structuring supramolecular gels" by Chivers and Smith, Nature Rev. Mat. 4, 463 - 478 (2019) should be included.
Response 1: We have added 6 references including the reference the reviewer 2 had suggested as follows.
1) Chivers, P. R. A.; Smith, D. K. Nature Rev. Mat. 2019, 4, 463-478. DOI: 10.1038/s41578-019-0111-6
2) Ruiz-Olles, J.; Slavik, P.; Whitelaw, N. K.: Smith, D. K. Angew. Chem., Int. Ed. 2019, 58, 4173-4178. DOI: 10.1002/anie.201810600 PubMed ID: 30682215
3) Yuan, T.; Xu, Y.; Fei, J.; Xue, H.; Li, X.; Wang, C.; Fytas, G.; Li, J. Angew. Chem., Int. Ed. 2019, 58, 11072-11077. DOI: 10.1002/anie.201903829
4) Hatai, J.; Schmuck, C. Acc. Chem. Res. 2019, 52, 1709-1720.DOI: 10.1021/acs.accounts.9b00142 PubMed ID: 31150198
5) Sinawang, G,; Kobayashi, Y.; Zheng, Y.; Takashima, Y.; Harada, A.; Yamaguchi, H.; Macromolecules, 2019, 52, 2932-2938. DOI: 10.1021/acs.macromol.8b02395
6) Tanaka, W.; Shigemitsu, H.; Fujisaku, T.; Kubota, R.; Minami, S.; Urayama, K.; Hamachi, I. J. Am. Chem. Soc. 2019, 141, 4997-5004. DOI: 10.1021/jacs.9b00715 PubMed ID: 30835456
Point 2: Although this work fulfills all the basic demands of a well conducted study, more details about the thermal stability and rheological properties of the gels should be included at least for some congeners.
Response 2: We have examined thermal stability of the gel from 4d and 4d/9 in chloroform. The gels from both of them collapsed at the same temperature ( ca. 45 °C), which was described in the text.
Point 3: The authors should also provide a probable explanation why the elemental analyses fail for some compounds (no satisfactory result was obtained).
Response 3: Compound 4b and 9 gave a satisfactory results in elemental analysis after drying them adequately and the compound 6d also gave good result after repeated recrystallization. These were described in the experimental section.
Round 2
Reviewer 1 Report
The manuscript has been improved and is ready for publication in the journal of "Gels". I hope the authors will report the applications of these gels in the future.